# Gain or Loss? Evidence for Legume Predisposition to Symbiotic Interactions with Rhizobia via Loss of Pathogen-Resistance-Related Gene Families

**DOI:** 10.3390/ijms232416003

**Published:** 2022-12-15

**Authors:** Katarzyna B. Czyż, Candy M. Taylor, Michał Kawaliło, Grzegorz Koczyk

**Affiliations:** 1Biometry and Bioinformatics Team, Institute of Plant Genetics Polish Academy of Science, 60-479 Poznań, Poland; 2School of Agriculture and Environment, The University of Western Australia, Perth, WA 6009, Australia

**Keywords:** Fabaceae, nodulation, nitrogen-fixing symbiosis, gene loss, comparative genomics, plant–pathogen interaction genes

## Abstract

Nodulation is a hallmark yet non-universal characteristic of legumes. It is unknown whether the mechanisms underlying nitrogen-fixing symbioses evolved within legumes and the broader nitrogen-fixing clade (NFC) repeatedly de novo or based on common ancestral pathways. Ten new transcriptomes representing members from the Cercidoideae and Caesalpinioideae subfamilies were supplemented with published omics data from 65 angiosperms, to investigate how gene content correlates with nodulation capacity within Fabaceae and the NFC. Orthogroup analysis categorized annotated genes into 64150 orthogroups, of which 19 were significantly differentially represented between nodulating versus non-nodulating NFC species and were most commonly absent in nodulating taxa. The distribution of six over-represented orthogroups within Viridiplantae representatives suggested that genomic evolution events causing gene family expansions, including whole-genome duplications (WGDs), were unlikely to have facilitated the development of stable symbioses within Fabaceae as a whole. Instead, an absence of representation of 13 orthogroups indicated that losses of genes involved in trichome development, defense and wounding responses were strongly associated with rhizobial symbiosis in legumes. This finding provides novel evidence of a lineage-specific predisposition for the evolution and/or stabilization of nodulation in Fabaceae, in which a loss of pathogen resistance genes may have allowed for stable mutualistic interactions with rhizobia.

## 1. Introduction

The legume family (Fabaceae) represents the third largest family of angiosperms with approximately 19,500 species distributed in diverse ecological and geographical habitats [1,2]. Traditionally, Fabaceae has been divided largely on the basis of floral morphological characters into three major groups: Caesalpinioideae, Mimosoideae and Papilionoideae. Recently, however, a new classification system based primarily on plastid *matK* gene sequences with support from a range of morphological traits was proposed (LPWG 2017) [3]. This revision has addressed key issues with the traditional classification system, including the non-monophyly of the Caesalpinioideae (LPWG 2013), and has now divided the family into six robustly supported monophyletic subfamilies, including (revised) Caesalpinioideae (148 genera), Cercidoideae (12 genera), Detarioideae (84 genera), Dialioideae (17 genera), Duparquetioideae (1 genus) and Papilionoideae (503 genera) [2,3].

Mutualistic symbiotic interaction with specialized gram-negative soil bacteria known as rhizobia is a hallmark trait of the Fabaceae family and confers a distinct adaptive advantage for plants by facilitating atmospheric nitrogen fixation within specialized root nodule structures. The trait is not uniformly spread throughout Fabaceae and is instead limited to its two most species-rich subfamilies. [3,4]. In particular, most of the surveyed species in Papilionoideae have the ability to symbiotically fix nitrogen, with only a few representatives among early diverging lineages of the subfamily lacking this trait [4]. While nodulation also occurs in Caesalpinioideae, it is confined only to a few genera within the subfamily. For example, the model genus *Chamaecrista* participates in rhizobial symbiosis, while closely related genera, such as *Senna*, are unable to participate [5].

To date, much effort has been put toward better understanding plant and rhizobia signaling exchanges and the mechanisms required for nodulation [6,7]. Nodule development and organogenesis-related loci have been characterized and divided into activities during the three overlapping stages of pre-infection, nodule initiation and differentiation [8]. It was evidenced that in the pre-infection stage, specific flavonoids released by legume roots serve as chemoattractants for the rhizobial symbiont and also activate expression of rhizobial *nod* genes [9]. Since then, rhizobial *nod* gene expression has been confirmed to result in lipooligosaccharide Nod factors perceived by a receptor in the legume host, triggering appropriate developmental responses. These include the curling of root hairs around the invading rhizobia, the entry of the rhizobia into the plant through infection threads and the induction of cell division in the root cortex, which marks the formation of the nodule [10,11].

The origin(s) of root nodulation in Fabaceae has similarly been a cause of great interest to researchers over the years, yet many uncertainties remain. Currently, one of the most widely accepted hypotheses for the origins of the trait in angiosperms more broadly is the ‘predisposition’ hypothesis in which a single evolutionary innovation is believed to have created a predisposition for symbiotic nitrogen fixation in the most recent common ancestor of the Nitrogen-Fixing Clade (NFC) in Rosids I plants [12,13]. In addition to legumes (Fabales order), the NFC contains a subset of nine other plant families from the Cucurbitales (Coriariaceae and Datiscaceae), Fagales (Betulaceae, Casuarinaceae and Myricaceae) and Rosales (Cannabaceae, Elaeagnaceae, Rhamnaceae and Rosaceae) orders, which all (with the exception of Cannabaceae) contain species that form root nodule symbiotic associations with actinorhizal gram-positive bacteria known as Frankia [14,15,16]. This predisposition for nodulation is believed to have been lost as many as roughly 17 separate times across different lineages of non-nodulating plants following divergence from the most recent common ancestor of the NFC [13]. In addition, the predisposition is thought to have led to an estimated number of eight independent gain-of-function events in nodulating lineages, after which nodulation (rather than the predisposition) was subsequently lost. The loss occurred approximately 10 times across different lineages, while further progression to a ‘stable fixer’ state (i.e., a state in which a lineage is very unlikely to lose the capacity for root nodule symbiosis) occurred in others [13]. The predisposition hypothesis means that, in addition to the multiple independent gains (of nodulation) among nodulating members of the NFC following the single predisposition innovation, it is possible that there were also multiple emergences of nodulation within the legume family itself [17]. 

While the origin(s) of nodulation and a predisposition towards its evolution (if it does indeed exist) are still unresolved, both individual gene gains or losses and whole-genome duplication (WGD) events have been considered as plausible scenarios leading to the development of this unique trait within various lineages of the NFC [17,18,19]. Since gene loss probably affects organisms to a greater extent than do most amino acid substitutions, it serves as one of the main drivers in the differentiation of gene families, morphological diversity, and adaptation, as well as in organogenesis and speciation [20,21,22]. Some members in one gene family may be lost in certain lineages, which may result, in extreme cases, in a deletion of the entire gene family and the creation of lineage-specific lost genes [23,24]. 

In our work, we have hypothesized that the differential distribution of nodulating species in some early-diverging legume taxa is reflected in convergent gene family changes throughout the clade. Moreover, we consider the possibility that differential gene loss, rather than the gain of several genes, together with a previously reported WGD might have been the driving force that contributed to genomic changes related to the evolution of persistent symbiotic nitrogen fixation in Fabaceae. Therefore, we have characterized gene families influenced by evolutionary pressure in legumes and several species from the NFC. Transcriptomic and genomic data from closely related genera of Caesalpinioideae legumes and so-called early diverging Papilionoideae have served as basal research material in order to better establish whether the gene family changes have been gradual or punctuated.

## 2. Results

### 2.1. Transcriptome Sequencing and Annotation

To complement existing legume omics resources, ten novel transcriptomes of species with varying capacities for symbiotic nitrogen fixation representing the Cercidoideae and Caesalpinioideae subfamilies were sequenced in order to better understand orthogroup differentiation between nodulating and non-nodulating legumes (sequencing data deposited in ArrayExpress under accession E-MTAB-11756). The length of the consensus assemblies varied more than two-fold between different species (Table 1), with *A. pechuelii* (Cercidoideae) emerging as the largest (398.9 Mb) and *F. albida* (Caesalpinioideae) representing the shortest transcriptome (140.0 Mb). The total transcript counts also varied considerably among the consensus assemblies, ranging from 42,219 for *S. obtusifolia* (Caesalpinioideae) to 63,658 for *C. siliqua* (Caesalpinioideae) (Table 1). 

As assessed through BUSCO land plant core ortholog annotations, the Evidential Gene approach for transcriptome assembly has proven to be the most successful in terms of ortholog completeness and redundancy reduction (number of duplicates), as well as maximization of coding potential (total length and completeness of encoded proteins, number of fragmented orthologs). Consistent with this finding, BUSCO analysis revealed an acceptably high (93.3–97.4%) ortholog completeness for all species using the EvidentialGene method (Table 1). The consensus merged approach of VELVET/OASES was similarly high when compared to the EvidentialGene approach in terms of completeness, but it introduced redundancy in the form of multiple false duplicated orthologs. In summmary, a single parameterization of assembly programs fails at capturing all classes of transcripts, highlighting the need for reconciliation of multiple approaches.

After BUSCO analysis, the transcriptomes were subjected to functional annotation after having been deemed to be of sufficient ortholog completeness and quality via BUSCO analysis. The number of reconstructed isoforms in each consensus transcriptome varied between 93,221 and 256,447 (Table 1). The percentage of duplicated orthologs ranged from 32.1% to 75.7%, with *C. mimosoides* and *C. sturtii* representing the least and most highly duplicated species in terms of apparent duplicates within the core Embryophyta ortholog set, respectively (Table 1). An average of 52% of transcripts were successfully annotated in each transcriptome. In general, the distribution of top gene ontology terms in the analyzed species was consistent between species (Appendix A). Interestingly, hydrolase activity gene ontology terms were overrepresented in *D. cinerea* and *P. stipulacea*. Both species are tropical trees (*D. cinerea*–Brazil, *P. stipulacea*–Africa), which raises the possibility that this trait may correlate with environmental adaptation to climate-specific niches. 

### 2.2. Differentially Represented Orthogroups in Nodulating and Non-Nodulating Species

Following the selection of representative transcripts for each putative locus in the 10 newly developed transcriptome assemblies, we utilized existing omics data for an additional 65 angiosperms (including 28 other Fabales) to delineate candidate gene families of common descent (i.e., orthogroups) using OrthoFinder. A total of 64,150 orthogroups were established from 2,245,917 of the 2,532,018 number of total genes within the Viridiplantae dataset (Table 2). Of these orthogroups, 45.2% were species-specific while 1.6% were represented in all 75 genomes/transcriptomes. All 64,150 orthogroups were tested for differential presence/absence and over-representation/under-representation type patterns in nodulating versus non-nodulating species across the entire Viridiplantae panel. 

Comparative presence/absence analysis identified nine orthogroups (OG0004107, OG0011465, OG0011581, OG0011714, OG0011803, OG0011851, OG0011900, OG0012127 and OG0012169) that differentiated nodulating from non-nodulating species (Table 3). In almost all cases, these orthogroups were more frequently absent in nodulating species. The only exception to this trend was OG0011581, annotated as 60S acidic ribosomal protein P0, which was contrastingly more commonly represented in nodulating species and absent in non-nodulating species. 

Comparative analysis of the over-representation/under-representation of orthogroups established 19 orthogroups with significantly different representation between nodulating and non-nodulating species (Table 4). As expected, this set included all nine orthogroups whose presence/absence significantly differentiated species capable of symbiotic nitrogen fixation from those that are not. Almost three quarters (13/19) of the differentially represented orthogroups (OG0004107, OG0010936, OG0011359, OG0011465, OG0011634, OG0011714, OG0011803, OG0011851, OG0011900, OG0012127, OG0012169, OG0012406 and OG0012477) were under-represented in nodulating species. Most genes within these groups were functionally annotated as being either involved in biotic and abiotic stress responses (including defense, wound and fungal disease responses); trichome development; growth regulation; the integration of cell shape and endoreplication levels; or in the dynamics of microtubules and organelle organization/assembly (Table 5). Only six orthogroups were over-represented in nodulating species (OG0000085, OG0001813, OG0002264, OG0009254, OG0011516 and OG0011581), which contained genes suggested to have roles in cell wall biogenesis/modifications; responses to toxic substances (including metabolic and catabolic responses to xenobiotic compounds); nodule development; and delayed aging (Table 5).

### 2.3. Phylogenetic Characterization of Orthogroups Under-Represented in Nodulating Species

The phylogenetic distribution of orthogroups with significant under-representation in nodulating species was closely examined to explore the timing of their evolution across our Viridiplantae dataset and their potential for facilitating the emergence and/or stabilization of symbiotic interactions, particularly within the Fabaceae family. These could be catalogued into three subsets (lost in legumes, lost in nodulating legumes and gradually lost irrespective of nodulation status).

Firstly, orthogroup and species phylogenetic tree reconciliation revealed the absence of three orthogroups in all legumes of both nodulating and non-nodulating status, including OG0011803 (Hapless protein), OG0012406 (Aspartyl protease family protein 2) and OG0012477 (Arabinogalactan protein 20). Each was common in the broader Viridiplantae panel and was represented in species distantly related to legumes, such as *O. sativa* and *A. thaliana*. In addition, it appeared each orthogroup was shared by the most recent common ancestor of the NFC, as their respective genes were identified in various members from Cucurbitales, Fagales and Rosales (Figure 1, Appendix A). However, the absence of genes from OG0012406 and OG0012477 in *P. lutea* and *Q. saponaria* suggests that these two particular orthogroups may have been lost in the most recent common ancestor of Fabales following divergence from its other relatives in the NFC. The presence of genes assigned to OG0011803 in *P. lutea* and *Q. saponaria* meanwhile indicated that this specific orthogroup was absent only from the Fabaceae family itself, and that its loss must have occurred at a later stage during the diversification of Fabales (Appendix A). 

Secondly, four under-represented orthogroups had been widely inherited across our representative Viridiplantae panel, including in all four orders within the NFC, but had been uniquely lost only in nodulating legumes. OG0011714 (Plant ubiquitin regulatory X domain (UBX) domain-containing protein 11), OG0012127 (Branchless trichome protein) and OG0012169 (TIFY, Jasmonate ZIM domain-containing protein) were present in one to two early Fabeaceae lineages (i.e., Detarioideae and/or Cercidoideae) but appeared to have been lost during later diversification of Fabaceae, with no representation observed beyond the *Umitza* clade of Caesalpinioideae in our set of analyzed taxa (Figure 1, Appendix A). The fourth orthogroup, OG0011900 (F-box protein GID2–gibberellin-insensitive dwarf protein 2), was similarly lost during the diversification of Caesalpinioideae, with genes from this orthogroup observed only in non-nodulating species from the *Umtiza* and *Cassia* clades (Figure 1, Appendix A). Neither the mimosoid clade (defined by Caesalpinioideae lineage divergence) nor Papilionoideae, which together contain almost all nodulating species from the legume family, had retained any genes from OG0011714, OG0011900, OG0012127 or OG0012169. The absence of representation of these orthogroups suggested that the losses of their associated genes, including those involved in trichome development and defense and wounding responses, were strongly associated with the evolution or stabilization of rhizobial symbiosis in legumes.

Lastly, the remaining six under-represented orthogroups appeared to have been gradually lost from Fabaceae independently of the gain/loss of nodulation within the family. OG0004107 (UDP-glucosyltransferase 86A2) was present at low frequency in almost all nodulating and non-nodulating taxa assessed from the Detarioideae, Cercidoideae and Caesalpinoideae subfamilies, with an average of 1.3 genes per species. However, the orthogroup was almost entirely absent from Papilionoideae (Figure 1, Appendix A). The exceptions to this included the non-nodulating diploid, *N. schottii* (two genes), and the nodulating allotetraploid, *A. hypogaea*. The latter is well-known to have experienced a WGD event and, not surprisingly, had the largest number (7) of OG0004107 genes among the legumes sampled in this study. OG0010936 (Transcription factor IBH1), OG0011359 (Transducin/WD40 repeat-like superfamily protein) and OG0011634 (unknown) also appeared to have been progressively lost during the diversification of the Papilionoideae subfamily (Figure 1, Appendix A). Meanwhile, OG0011465 (F-box protein CPR1-like, constitutive expresser of PR genes 1) and OG0011851 (Kinesin-like protein, KIN-8B) appeared to have been lost at an earlier stage within the family’s diversification. Specifically, one to two genes in species from the *Cassia*, *Senna* and *Chamaecrista* genera were present, yet their absence thereafter indicated a punctuated loss of the orthogroups in Caesalpinioideae sometime after the divergence of the *Cassia* clade and prior to the divergence of the mimosoid clade (Appendix A). 

### 2.4. Phylogenetic Characterization of Orthogroups Over-Represented in Nodulating Species

Next, we similarly investigated the phylogenetic distribution of the six orthogroups that were significantly over-represented in nodulating species to decipher whether they may have a clear association with the emergence and/or stabilization of nodulation within the NFC and/or legume family.

The first subset comprised ‘gained’ orthogroups that were exclusively represented in Fabales and were absent from all other members of the Viridiplantae dataset, including actinorhizal taxa. These encompassed orthogroups OG0001813 (Fasciclin-like arabinogalactan protein 11), OG0009254 (Embryonic abundant protein universal stress protein 92 (USP92)), OG0011516 (Dirigent protein 21) and OG0011581 (60S acidic ribosomal protein P0). All four orthogroups were frequently observed among early (e.g., the ADA clade and Cladrastideae, Dalbergieae and Genisteae tribes) to later evolved Papilionoideae lineages (e.g., the NPAAA clade), with an average of 1.3 (OG0011581) to 2.7 (OG0009254) genes per Papilionoid species. However, each orthogroup appeared to have first emerged at an earlier stage of diversification within the legume family, as their associated genes were identified in species from one or more of the *Umtiza*, *Cassia* and mimosoid clades of Caesalpinioideae (Figure 1, Appendix A). As this distribution also comprises non-nodulating species, it indicates that all four orthogroups are not exclusive to nodulating species.

The remaining two orthogroups were contrastingly present in the broader Viridiplantae panel but were increasingly over-represented in Fabales, particularly within nodulating members of Fabaceae. The first candidate family, OG0000085 (UPD-glucosyltransferase protein) was well-represented in all 75 genomes/transcriptomes assessed in the study. However, its average gene count rose approximately 1.5- and 2-fold from 9.7 genes in species excluding the Fabales order to 14.1 and 19.3 genes per species amongst non-nodulating and nodulating Fabales taxa, respectively (Figure 1, Appendix A). The other orthogroup, OG0002264 (Soyasapogenol B glucuronide galactosyltransferase) was meanwhile represented by a single gene copy in *Manihot esculenta* outside of the NFC. The frequency at which the orthogroup was represented increased slightly among the wider NFC though, with its associated genes observed in a third of species from Cucurbitales, Fagales and Rosales (Appendix A). However, OG0002264 experienced a drastically large expansion during the later diversification of Fabaceae, with an average of 5.3 genes per nodulating member of the family. The expansion was particularly evident in Papilionoideae, where the orthogroup was represented in all species and a maximum of 29 members were observed in *M. truncatula* (Appendix A). 

### 2.5. Case Study Validation of In-Silico Estimations of Gene Content

Seven orthogroups were selected for further examination using experimental approaches. The intent of this work was to: (i) verify the accuracy of our in silico approach for the estimation of gene copy number variation; (ii) establish gene expression variation for species where genome resources were originally used for orthogroup analysis; and (iii) gain further insight into duplication events that lead to the expansion of select orthogroups in different plant lineages. 

In our Viridiplantae dataset, a grand total of 84 genes encoding homologs of Embryonic abundant protein USP92 were assigned to OG0009254. While the occurrence of these homologous genes was indeed largely restricted to nodulating species, there were three exceptions to this trend (*Ceratonia siliqua*, *Nissolia schottii* and *Senna hebecarpa*). Consequently, no clear trend could be ascribed to the distribution of this orthogroup among legumes. In some lineages, there was clear evidence of duplication within genome regions carrying OG0009254 representatives (Figure 1, Appendix A). This duplication, however, did not appear to have a common ancestral origin and was instead lineage-specific, impacting only a limited few species in the panel, such as *Arachis hypogaea*, *Astragalus membranaceus*, *Cicer arietinum*, *Trifolium pratense* and *Medicago truncatula.* Furthermore, even within single lineages, there was evidence that duplication of OG0009254 may have occurred at a species-specific level. For example, while *Glycine max* has six *USP92* orthologues, transcriptomic analysis of its close relative *G. soja* revealed only two gene copies. Taking together gene expression analysis and copy number estimations from in silico and ddPCR, we hypothesize that OG0009254, despite being highly duplicated in several species, has just two or three dominant variants expressed actively within the legume family. We established that in the *G. max* genome only GLYMA_12G217300, GLYMA_13G283900 and GLYMA_12G217400 sequences were transcribed. Additional representatives (GLYMA_08G230600, GLYMA_08G036600 and GLYMA_08G036700) could be pseudogenes or exhibit organ specific patterns of expression (Appendix A). Similar trends in expression profiling of identified genes were observed in *M. truncatula*. 

The second over-represented orthogroup we analyzed was OG0011581, (60S acidic ribosomal protein P0); the only orthogroup which was also strongly indicated in the presence/absence testing. There were 48 representative orthologous genes within this group, all restricted only to Fabaceae (Figure 1, Table 5). While these genes were found in almost all nodulating species, they were also transcribed in a comparatively smaller proportion of non-nodulating taxa, including *C. siliqua*, *S. hebecarpa*, *C. sturtii*, *S. obtusifolia* and *N. schottii*. The ancient WGD in the ancestor genome of all Papilionoideae (referred to as WGD6 by Zhao et al. 2021) may in this case be the cause of over-representation in nodulating species, as 10 of the 26 species belonging to this subfamily and from various clades within it (e.g., genistoid, dalbergioid and NPAAA clades) have duplicated or triplicated 60S acidic ribosomal protein P0 gene sets [18]. Meanwhile, all Caesalpiniaceae representatives of the orthogroup were found to possess only single gene copies (Figure 1, Appendix A). Within our 10 transcriptomes, both *A. pechuelii* and *G. dioicus* lacked these genes. The expression of identified gene variants was confirmed in: *C. mimosoides, C. siliqua, C. sturtii, D. cinerea, D. velutinus, F. albida, P. stipulacea* and *S. obtusifolia*, as well as in model legumes: *C. cajan*, *C. arietinum*, *G. max*, *L. japonicus*, *L. angustifolius, M. truncatula* and *V. radiata* (Appendix A). Nevertheless, we have noticed some alterations in gene copy number between putative transcriptome loci and ddPCR results. We observed no ddPCR signal for sequences Cstu018553t1 (567 bp), Csili017540t1 (582 bp), Cmim018396t1 (570 bp) and Pstip016651t1 (564 bp) even though they were verified as being expressed in those species (Appendix A). Apart from de novo assembly artifacts, it is important to note that the separation of positive and negative fluorescence during droplet digital PCR might have been affected by the total load of DNA, including primer concentrations. Moreover, the inaccurate designation of restriction enzyme sites and incomplete digestion of genomic DNA could interfere with the results of copy variants estimations for both species.

Lastly, we experimentally confirmed the orthologue copy number from OG0004107, OG0011465, OG0011851, OG0011900 and OG0012127 identified in our newly sequenced transcriptome dataset, which verified that our in silico predictions were accurate (Appendix A). This confirmation provided additional confidence in the phylogenetic patterns described above, in which these gene families were under-represented in nodulating species and gradually lost from legume genomes. 

Taken together, inspection of the integrated OrthoFinder species and gene trees with experimental investigation of gene copy number/expression illustrated that although the comparative statistical analyses between nodulating and non-nodulating species were significant, there was no clear distribution pattern for the over-represented orthogroups within the legume family. The extant state of the six overrepresented orthogroups thus implies that genomic evolution events tied to gene family expansion, including ancestral WGDs, were not the likely scenarios directly contributing to the development of stable symbiotic interactions within the family as a whole.

## 3. Discussion

A small fraction of angiosperms are able to supplement their nitrogen uptake from the soil by forming mutualistic symbiotic associations with rhizobia and *Frankia* bacteria that colonize specialized structures and convert atmospheric nitrogen into ammonium. As the ability to better resolve taxonomic relationships among angiosperms previously inferred from morphological characters has enhanced over the years, so too has our collective understanding of the origins of root nodulation in these plants and their unique and highly valuable capacity for symbiotic nitrogen fixation. For example, phylogenetic analysis of the chloroplastic *rbcL* gene led to the undisputed discovery that all nodulating angiosperms belong to four orders (Cucurbitales, Fabaceae, Fagales and Rosales) within a larger monophyletic clade of Rosids I, termed the NFC [12]. 

Clarity on the evolution of this trait will continue to accelerate as recognition is gained for the need for accurately curated databases of angiosperm nodulation status [13,25,26,27] and next-generation sequencing-based genotyping continues to rapidly increase in affordability [28]. However, forming meaningful conclusions from comparative and phylogenomic analyses heavily relies on researchers being able to utilize omics resources for a large number of plants, both within and outside of the NFC and of nodulating and non-nodulating status. To that end, we sequenced the transcriptomes of 10 legume species belonging to basal clades within the family. Nine of these species have not previously had such resources, such as those available for model and/or agriculturally important legumes (particularly from the Papilionoideae subfamily). Therefore, these constitute useful additions to the growing collection of assemblies for less well-resourced plants, such as those generated through large-scale collaborations like the One Thousand Plant Transcriptomes Initiative (2019) [29]. 

Currently, one of the models best thought to explain the distribution of nitrogen-fixing symbiosis trait in flowering plants predicted a single predisposition event at the base of the NFC followed by as many as 10 independent origins [30]. The wide acceptance of this theory is based on it: (i) being a parsimonious scenario supported by phylogenetic modelling (e.g., [13]); and (ii) accounting for the variation observed in microsymbionts within and between NFC lineages, nodule physiology and nodule ontogenesis [31]. Two other general hypotheses have also been considered by the scientific community. The first of these is the ‘multiple origins’ hypothesis, in which there was convergent de novo evolution of nodulation via multiple independent gains in different lineages without a prior common predisposition for the trait. However, the genetic complexity of nodulation and the fact that there are many common features present across divergent nodulating lineages that are all limited to a single angiosperm clade deemphasizes this scenario. The second alternative hypothesis is the ‘single origin’ hypothesis, where nodulation evolved in a recent common ancestor of the NFC and was subsequently lost multiple independent times in non-nodulating lineages [31]. Although it has been critiqued for not being a parsimonious or likely scenario due to the incredibly large number of independent losses of function that would be required to explain the relatively small proportion and distribution of nodulating species within the NFC [30], independent studies have recently been gathering genomic-based evidence supporting this hypothesis, such as discovering the loss or pseudogenization of genes indispensable for root nodule symbiosis (such as *NODULE INCEPTION, NIN,* and *NOD FACTOR PERCEPTION*, *NFP*) from non-nodulating species within the NFC that otherwise appear present in rhizobial and actinorhizal symbionts [19,32]. In addition, phylogenomic analyses suggest that the lineage-specific mutation of critical genes like *NIN* may have facilitated a switch from actinorhizal to rhizobial microsymbionts in Fabaceae [18]. The acquisition of such a mutation could explain the diversity in various nodulation characteristics throughout the NFC under the single origin hypothesis. 

If nitrogen-fixing symbiosis evolved multiple times, particularly in parallel from a pre-disposed state common to the NFC, expansions in the same gene families among legumes and other nodulators would be likely. Using our dataset, we searched for gene families with statistically significant differences in representation between nodulating and non-nodulating species to assess this assumption but did not find evidence to support this theory. No characteristic features for all nodulating species in the NFC were observed that would implicate parallel or convergent recruitment of nitrogen-fixing symbiosis. Instead, we observed lineage-specific over-representation and even cases of exclusive representation of six gene families in Fabaceae (Figure 1, Table 4). The biological roles of some of these genes in response to toxic and xenobiotic substances, translation and cell wall biogenesis (Table 5, Appendix A) suggests they could have plausible involvement in the evolution of rhizobial symbiosis. Nevertheless, the over-representation of some families, such as *USP92* (OG0009254), reflects well-documented WGD duplication events within the family, especially for domesticated pulse crops within the Papilionoideae subfamily [17,18]. Therefore, these five characterized gene families could simply just be signatures of legume genome evolution or alternatively support the hypothesis that an ancestral Papilionoideae polyploidy event led to the stabilization of nodule symbiosis in this subfamily once the trait had already emerged [28]. Determining whether there was similar parallel expansion of these five gene families in Cannabaceae, the only other lineage with rhizobial symbiosis and which was only represented by a single non-nodulating species in our current dataset, may offer further insight into the likelihood of each of these three possibilities. 

Irrespective of how symbiotic nitrogen fixation via root nodulation originated within the NFC, it is abundantly clear that multiple losses of the trait have occurred throughout the clade that have impacted its current distribution among extant nodulating lineages. Although we did not find similar results in our study, perhaps because not all genes involved in the trait may have been actively expressed in the leaf-derived transcriptomes of nodulating species that we utilized for our comparative analysis, there is evidence that loss of the trait is directly related to the parallel loss or pseudogenization of genes critical for nodule organogenesis [18,19,31]. An equally important question stemming from these massive losses that is yet to be answered is: what characteristics make some lineages, such as Papilionoideae, less vulnerable to the loss of nodulation than others?

As discussed earlier, we found evidence for possible expansions of six gene families of nodulating legumes that may represent genomic changes that formed a family-specific predisposition for root nodule symbiosis or alternatively led to the stabilization of the trait within the family. In addition to this finding, however, we also detected significant under-representation, and even the loss in some cases, of 13 gene families in nodulating legumes and the wider Fabaceae (Figure 1, Table 4). Notably, this set includes genes with putative roles in wounding and general plant defense responses, plus the regulation of growth, phytohormone signaling and trichome morphogenesis which also influence biotic stress responses [33]. The loss of such genes would potentially therefore incite less-hostile reactions to rhizobia and promote a beneficial interaction between organisms. Recent studies have suggested that jasmonate ZIM (zinc-finger inflorescence meristem) domain proteins (JAZ) (orthogroup OG0012169) in the jasmonate (JA) signaling pathway may be involved in the generation of a response to plant pathogen attacks. Tomato JAZ proteins regulate the progression of cell death during host and nonhost interactions [33]. It was also documented that aspartyl proteases (AP) (OG0012406) are directly involved in pathogen resistance. Pathogenesis-related (PR) proteins secreted upon pathogen challenge were degraded by an extracellular aspartic protease, preventing its over accumulation. An aspartic protease gene detected in tomato leaves (LeAspP), in response to wounding and treatments within the system and methyl jasmonate (MeJA), was also shown to be systemically induced, suggesting that this AP plays a role in defense against pathogens [34]. Recently, arabinogalactan proteins (AGP) (OG0012477) have been indicated to play key roles at various levels of interaction between roots and soil-borne microbes, either beneficial or pathogenic [35]. AGPs were described to be involved in attracting and initiating root tip colonization by beneficial microbes. They were also found to be expressed at the interface of infectious structures that are formed between various beneficial microbes and root cells, and which allow the exchange of nutrients between the root and its symbiont. Interestingly, in a pathogenesis context, they are also likely to set the scene for mounting an efficient and localized defense response. Based on recent findings of their antimicrobial properties, AGPs are directly involved in controlling some pathogenic microbes [35]. Taking together plausible scenarios leading to the development of a predisposition to nodulation, we hypothesize that the change of rhizosphere environment by secreting different sets of proteins could attract and enable different soil-bacteria to interact with host plants. Such an interaction could influence and enable the evolution of nitrogen-fixing symbiosis. Last but not least, a gradual loss of the gene family encoding branchless trichome (BLT) (OG0012127) could indirectly influence the development of the predisposition to nodulation via modulation of branch sites, trichome cell shape and integrating endoreplication levels with cell shape. 

While both the expansion and loss of selected gene families may be important, the sheer number of under-represented gene families relative to those with over-representation suggests to us that a parallel loss of plant-pathogen defense response genes could have predisposed to the initial evolution or stable persistence thereafter of symbiotic interactions in nodulating Fabaceae lineages. We postulate that this predisposition for nodule evolution or stabilization, if indeed real, would be unique to the legume family and not shared by another NFC order in the event that they too were found in possession of a similar predisposition or stabilizing genetic signature for two key reasons. Firstly, our phylogenetic analysis indicated the 13 gene families under-represented in legumes were abundant in the broader angiosperm panel and indeed present in the genomes of nodulating species from Cucurbitales, Fagales and Rosales (Figure 1). The losses occurred following the divergence of a recent common ancestor of Fabales from the NFC, or at an even later stage of diversification within the Fabaceae. Secondly, the astounding biogeographical and ecological success of the legume family suggests that its two most species-rich subfamilies, which are also the only ones capable of root nodulation, had a trait and genetic background that was favorable for colonizing new, limiting and niche environments. Supplementary nitrogen resources would certainly pose a fitness advantage for the extensive radiation of Fabaceae, which has not been matched by the other nine plant families partaking in symbiotic nitrogen fixation.

## 4. Material and Methods

### 4.1. Research Material

The following germplasm was obtained from (i) U.S. National Plant Germplasm System: *Cicer arietinum* (LINE 6560) and *Gymnocladus dioicus* (Ames 2917); (ii) Royal Botanic Gardens, Kew: *Adenolobus pechuelii* (82,389), *Cassia sturtii* (209,803), *Ceratonia siliqua* (165,598), *Chamaecrista mimosoides* (81,072), *Desmanthus velutinus* (326,922), *Dichrostachys cinerea* (171,155), *Faidherbia albida* (104,805), *Medicago truncatula* (859,574), *Mimosa pudica* (19,956), *Piptadenia stipulacea* (102,797) and *Senna obtusifolia* (92,737); (iii) IPK Gatersleben: *Cajanus cajan* (CAJ2), *Glycine max* (SOJA 1333), *Phaseolus vulgaris* (PHA 6055) and *Vigna radiata* (VIG 1525); and (iv) IPG PAS: *Lotus japonicus* and *Lupinus angustifolius* (Sonet). The seeds were germinated in Petri dishes at temperatures optimal for each species (varying from 22 °C to 28 °C for Mediterranean vs. tropical species) and transferred to pots in a cultivation room (16 h/8 h photoperiod; 22 °C constant temperature; 70–80% humidity). No symbiotic bacteria were added to the standard potting mix soil.

### 4.2. RNA and DNA Isolation

RNA isolation from combined leaf and stem tissues was carried out using a Spectrum Plant Total RNA Kit (Sigma Aldrich) with minor modifications to the manufacturer’s protocol. Optimization of the protocol included: (i) decreasing the amount of ground tissue used for extraction (50 mg instead of the proposed 100 mg); (ii) incubating samples for 5 min at 56 °C during lysis; and (iii) using 750 μL of binding solution. DNase digestion was incorporated during RNA isolation using an On-Column DNase I Digestion Set (Sigma Aldrich, St. Louis, MO, USA). RNA concentration and quality was measured using an Experion Automated Electrophoresis System with an RNA StdSens Analysis Kit (Bio-Rad, Hercules, CA, USA).

High-quality DNA was isolated using a DNeasy Plant Pro Kit (Qiagen, Hilden, Germany) without changes to the manufacturer’s protocol. DNA yield was estimated by Nanodrop (ThermoFisher, Waltham, MA, USA) and quality assessed by electrophoresis on a 2% agarose gel.

### 4.3. RNA Sequencing, Transcriptome Assembly and Annotation

RNA from three biological replicates was sequenced for ten non-model legume species, including: *A. pechuelii*, *C.siliqua*, *C. mimosoides*, *C. sturii*, *D. velutinus*, *D. cinerea*, *F. albida*, *G. dioicus*, *P. stipulacea* and *S. obtusifolia*. The RNA libraries were prepared using TruSeq RNA Stranded mRNA and were sequenced on an Illumina Nova Seq 6000 platform using a pair-end read (2 × 151 bp) approach (Macrogen, Amsterdam, The Netherlands).

A combined strategy of multiple merged assemblies [36] was deployed to obtain the best coverage possible of different groups of putative transcripts. Firstly, the obtained raw data were filtered to remove low-quality reads (<Q10) before having primers, adaptors and low-quality terminal residues trimmed using bbmap (version 38.79). The pre-processed high-quality paired reads were assembled de novo using: (i) Trinity 2.5.1; (ii) SOAPdenovo-TRANS 1.04 (k-mer length of 31–71 bp; step of 10 bp) with additional gap finishing by GapCloser (version 1.12); (iii) VELVET/OASES (respective version: 1.2.10 and 0.2.09) pipeline at multiple values of k-mer length (20–70 bp). The individual assemblies were merged using the EvidentialGene toolkit [37] (downloaded on 10 December 2019). Ortholog completeness of the resulting curated assemblies was assessed using BUSCO v3 [38] with the Embryophyta reference set (embryophyta_odb9). The assembled transcripts were functionally annotated using Blast2GO Pro 5.2 vs. UniProt/SwissProt, as well as using the EnTAP 0.9.2 automated annotation pipeline [39]. Snakemake 5.19.3 [40] workflow was used to control the execution of all steps of quality control, reconstruction and annotation. RSEM 1.3.2 was used to estimate expression levels for comparisons with real-time expression profiling.

### 4.4. Identification of Gene Families across Nodulating and Non-Nodulating Species

OrthoFinder 2.4.0 [41] was used to infer orthogroups (i.e., groups of genes descended from a single gene in the last common ancestor of the analyzed species) based on representative protein sequences from each unique putative locus. These representative sequences included canonical isoforms in the case of genomic data or the longest coding isoform of a locus in the case of transcriptomic data. As OrthoFinder employs Markov clustering, each representative gene sequence was exclusively assigned to a single orthogroup. OrthoFinder was also used to generate the species tree used to visualize the changes in representation. 

A total of 74 species were used in the OrthoFinder analysis (Appendix A), including two monocot species (*Brachypodium distachyon* and *Oryza sativa* ssp. *japonica*), which were included to provide a suitable outgroup. The individual datasets included both transcriptomes (newly reconstructed herein and reference data from the One Thousand Plant Transcriptome Initiative (2019)), as well as model genomes. Two independent transcriptomes, including one assembled in this study, were analyzed for *G. dioicus*. The capacity of each species to nodulate was determined from a consensus survey of five comprehensive phylogenetic studies and databases [13,18,25,26,27] (Appendix A).

Functional annotation of orthogroups that differentiated nodulating and non-nodulating species (see Section 2.5) was conducted via Blast2Go, literature review, a search of Gene Ontology (GO) terms associated with *A. thaliana* and *G. max* orthologues within the UniProtKB database, as well as TAIR/GOSLIM, and with EnTAP against our newly sequenced transcriptomes.

### 4.5. Statistical Testing of Differential Representation of Gene Families

Statistical testing was carried out with the statistics (stats) module of SciPy toolkit (version 1.2.1). The identified orthogroups were contrasted between nodulating and non-nodulating species by conducting orthogroup absence/presence (via Fisher’s exact tests) and normalized count (via pairwise, independent samples, *t* tests) comparisons. For the latter, individual species counts of genes per orthogroup were normalized against their respective total genome/transcriptome gene numbers to account for different genome duplication histories (Appendix A). In both cases, Bonferroni correction was employed to correct for multiple testing, with a significance threshold of 0.001.

### 4.6. Estimating Sequence Variant Numbers of Selected Orthogroups

To experimentally verify our in-silico estimates of gene copy numbers in selected orthogroups that were differentially represented in the genomes of nodulating and non-nodulating species, droplet digital PCR (ddPCR) was performed with the use of the Bio-Rad QX200 Droplet Digital PCR System (Bio-Rad). Sets of species-specific primers (Appendix A) were designed to cover each reported gene family member. Due to it being a single-copy gene in all 9 novel legumes sequenced in this study plus all additional model legume species within the assembled Viridiplantae panel, *Phytolongin Phyl1.1* (*PHYL1.1*) was used as a reference gene for normalization (Appendix A). All ddPCR reactions contained 5 ng DNA, 2× QX200 ddPCR EvaGreen Supermix (Bio-Rad) and 1 µM gene-specific primers. The final volumes of ddPCR reactions (20 μL), together with 70 μL of droplet generation oil, were placed in DG8 Cartridges, partitioned into droplets by the QX200 Droplet Generator (Bio-Rad, Hercules, CA, USA) and transferred into 96-well plates. The ddPCR thermal cycling protocol involved initial denaturation (95 °C for 5 min), followed by 40 cycles of: denaturation (95 °C, 30 s), annealing (60 and 61 °C, 30 s), elongation (72 °C, 45 s) and final elongation (72 °C, 45 s). Fluorescence was measured on the QX200 Droplet Reader (Bio-Rad, Hercules, CA, USA). On average, 17,000 droplets were analyzed per 20 μL ddPCR. Data analysis was performed with QuantaSoft droplet reader software (Bio-Rad, Hercules, CA, USA).

### 4.7. Expression Analysis

Quantitative PCR (qPCR) was performed and analyzed using a C1000 Thermal Cycler CFX 96 Real-Time System (BioRad, Hercules, CA, USA) with SYBR green fluorescent dye detection. Three biological and three technical replicates for each species were subjected to analysis. Two reference housekeeping genes, *Helicase* (*HEL*) and DNA damage-repair/toleration protein (*DRT102*), were amplified for each RNA template. Each qPCR reaction contained 20 ng of RNA, 300 nmoles each of forward and reverse primers, 1× iTaq PCR reaction mix, 1× iScript reverse transcriptase (BioRad, Hercules, CA, USA) and nuclease-free water to give a total volume of 10 µL. No Reverse Transcription (NRT) and No Template Controls (NTC) were included for each RNA template. The thermal cycling program comprised an initial stage for reverse transcription at 50 °C for 10 min, and a second stage for qPCR featuring initial denaturation at 95 °C for 1 min, followed by 40 cycles of denaturation at 95 °C for 10 s and primer annealing and extension at 61 °C for 30 s. Melt-curve analyses were then conducted before concluding the program with a 4 °C hold.

## 5. Conclusions

We assembled ten newly sequenced transcriptomes representing species from the Cercidoideae and Caesalpinioideae legume subfamilies, which when supplemented with omics data from 65 angiosperms, were used to better understand the relationship between gene content and the evolution of nodulation in legumes and the wider NFC via orthogroup analysis. We found an expansion of several gene families was limited to Fabaceae, with no differentiation between nodulating and non-nodulating species. Gene gains (e.g., through ancestral WGDs) therefore do not appear to have directly influenced the origin(s) or generally hypothesized predisposition for symbiotic interaction within the NFC. Instead, we observed that gradual or punctuated losses of several gene families within Fabaceae influenced the genetic background of nodulating lineages and are tied to stable nitrogen-fixing symbiosis with rhizobia. The losses of genes involved in plant–pathogen interactions were particularly striking and could have contributed to improved compatibility between legumes and rhizobia, thereby stabilizing mutualistic symbiosis between the two organisms. The transcriptomes developed in this study will serve as useful resources for comparative genomics studies to further explore the evolution of nodulation and other traits in legumes.

## Figures and Tables

**Figure 1 ijms-23-16003-f001:**
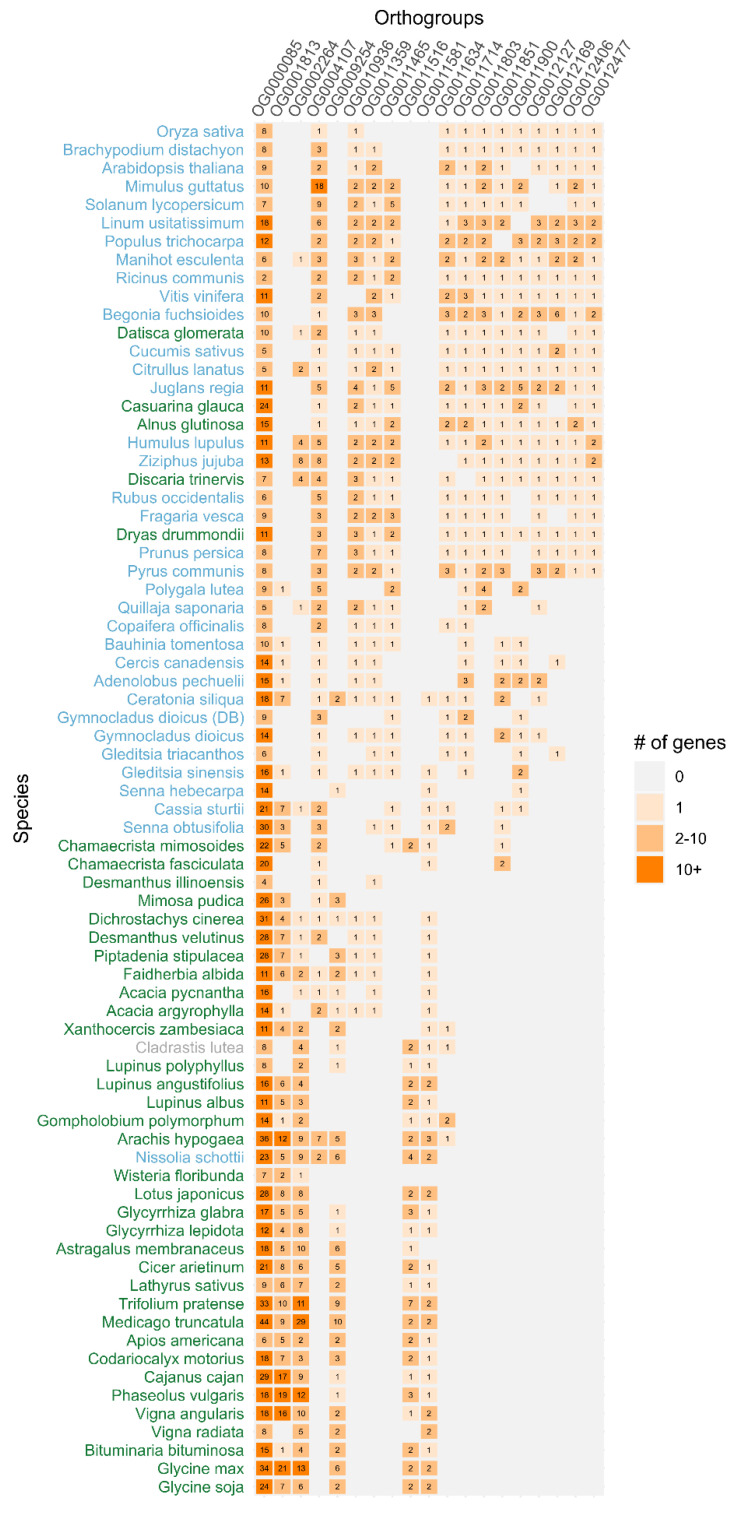
Representation of all genes counts for 19 orthogroups across 74 analyzed species. Font color reflects the status of nodulation for each species: green—nodulating species, blue—non nodulating species and grey—no status assigned.

**Table 1 ijms-23-16003-t001:** Transcriptome assembly statistics.

Species Name	Transcripts (nb)	Isoforms (nb)	Duplicated Orthologs (%)	Total Length (bp)	Ortholog Completeness (BUSCO%)	Annotated Transcripts (%)	Average Seq Length (bp)	Longest Seq (bp)	Shortest Seq (bp)
*A. pechuelii*	55,331	211,746	63.2	398,942,241	95.2	56.5	881	16,116	132
*C. mimosoides*	43,838	99,644	32.1	152,146,755	96.3	59.1	856	16,431	129
*C. siliqua*	63,658	169,042	48.2	277,707,898	94.3	49.9	722	15,342	132
*C. sturtii*	63,247	256,447	75.7	359,052,434	97.4	44.8	708	16,479	129
*D. cinerea*	58,814	202,614	72.9	274,297,199	95.5	43.9	698	15,309	117
*D. velutinus*	50,142	130,304	51.2	189,491,013	95.5	49.5	774	16,482	132
*F. albida*	46,192	93,308	38.5	140,099,747	94.7	51.5	780	15,303	132
*G. dioicus*	54,658	148,138	43.9	255,768,827	95.8	52.7	787	16,011	132
*P. stipulacea*	43,370	108,465	51	161,251,853	93.3	54.5	806	16,336	132
*S. obtusifolia*	42,219	93,221	40	146,753,099	95.1	58.2	853	16,482	129

**Table 2 ijms-23-16003-t002:** Summary of OrthoFinder analysis to identify orthologous groups of genes from the transcriptomes and genomes of 75 Viridiplantae plant species.

OrthoFinder Statistic	Value(s)
Total number of annotated genes	253,2018
Number (and percentage) of genes assigned to orthogroups	2,245,917 (88.7%)
Total number of orthogroups	64,150
Number (and percentage) of species-specific orthogroups	29,012 (45.2%)
Number (and percentage) of genes in species-specific orthogroups	140,274 (5.5%)
Number (and percentage) of orthogroups universal to all species	1069 (1.6%)
Number (and percentage) of single gene-copy orthogroups	0 (0%)
Mean number of genes per orthogroup	35.0
Median number of genes per orthogroup	3.0

**Table 3 ijms-23-16003-t003:** Gene families differentially represented in nodulating and non-nodulating species. Presence/absence analysis, Fisher Exact Test with Bonferroni correction. Dataset column indicates the mode of comparison: Fabales denotes nodulating Fabales vs. all other taxa (including non-nodulating legumes), Viridiplantae denotes all known nodulators among green plants vs. all non-nodulators.

Orthogroup	Nodulating Species		Non-Nodulating Species		Odds Ratio	*p*-Value	Dataset
	Present	Absent	Present	Absent			Fabales
OG0004107	11	24	38	1	0.012	0.000115422	Fabales
OG0011465	2	33	30	9	0.018	2.06 × 10^−5^	Fabales
OG0011581	30	5	6	33	33	0.000144653	Viridiplantae
OG0011714	4	35	31	4	0.014	5.17355 × 10^−7^	Fabales
OG0011803	1	34	26	13	0.014	0.000466401	Fabales
OG0011851	3	32	30	9	0.028	0.000175831	Fabales
OG0011900	1	34	28	11	0.011	3.15883 × 10^−5^	Fabales
OG0012127	1	34	26	13	0.014	0.000466401	Fabales
OG0012169	0	35	23	16	0	0.000599504	Fabales

**Table 4 ijms-23-16003-t004:** Gene families differentially represented in nodulating and non-nodulating species. Student’s t-test for normalized gene counts. Dataset column indicates the mode of comparison: Fabales denotes nodulating Fabales vs. all other taxa (including non-nodulating legumes), Viridiplantae denotes all known nodulators among green plants vs. all non-nodulators.

Orthogroup	Representation in Nodulating Species	T-Stat	*p*-Value	Dataset
OG0000085	Over-represented	6.793068028	0.00034066	Fabales
OG0001813	Over-represented	6.708229518	0.000487269	Fabales
OG0002264	Over-represented	6.544150538	0.000971223	Fabales
OG0004107	Under-represented	−8.474327743	2.5597 × 10^−7^	Viridiplantae
OG0009254	Over-represented	7.471962066	1.89568 × 10^−5^	Fabales
OG0010936	Under-represented	−6.827187958	0.000294919	Fabales
OG0011359	Under-represented	−6.731102129	0.000442482	Fabales
OG0011465	Under-represented	−8.697111523	9.82762 × 10^−8^	Viridiplantae
OG0011516	Over-represented	7.333314606	3.42939 × 10^−5^	Fabales
OG0011581	Over-represented	8.602542801	1.47528 × 10^−7^	Fabales
OG0011634	Under-represented	−7.366384859	2.97753 × 10^−5^	Fabales
OG0011714	Under-represented	−10.61360496	2.87832 × 10^−11^	Viridiplantae
OG0011803	Under-represented	−7.434134842	2.22871 × 10^−5^	Fabales
OG0011851	Under-represented	−8.049506863	1.59019 × 10^−6^	Fabales
OG0011900	Under-represented	−8.312467416	5.13357 × 10^−7^	Fabales
OG0012127	Under-represented	−7.493432602	1.72924 × 10^−5^	Fabales
OG0012169	Under-represented	−6.921957525	0.000197467	Fabales
OG0012406	Under-represented	-6.646016389	0.000633175	Fabales
OG0012477	Under-represented	−6.646465092	0.000631981	Fabales

**Table 5 ijms-23-16003-t005:** Differentially represented orthogroups annotation and final counts.

Orthogroup	Representation in Nodulating Species	Total Number (Viridiplantae Dataset)	Functional Annotation (Blastp)
OG0000085	Over-represented	1119	UDP-glucosyltransferase protein
OG0001813	Over-represented	238	Fasciclin-like arabinogalactan protein 11
OG0002264	Over-represented	212	Soyasapogenol B glucuronide galactosyltransferase
OG0004107	Under-represented	143	UDP-glucosyltransferase 86A2
OG0009254	Over-represented	84	Embryonic abundant protein USP92
OG0010936	Under-represented	62	Transcription factor IBH1
OG0011359	Under-represented	53	Transducin/WD40 repeat-like superfamily protein
OG0011465	Under-represented	51	F-box protein CPR1-like
OG0011516	Over-represented	50	Dirigent protein 21
OG0011581	Over-represented	48	60S acidic ribosomal protein P0
OG0011634	Under-represented	47	unknown
OG0011714	Under-represented	45	Plant UBX domain-containing protein 11
OG0011803	Under-represented	43	Hapless protein
OG0011851	Under-represented	42	Kinesin-like protein KIN-8B
OG0011900	Under-represented	41	F-box protein GID2
OG0012127	Under-represented	36	Branchless trichome protein
OG0012169	Under-represented	35	TIFY, Jasmonate ZIM domain-containing protein
OG0012406	Under-represented	30	Aspartyl protease family protein 2
OG0012477	Under-represented	30	Arabinogalactan protein 20

## Data Availability

All sequencing data concerning ten novel transcriptomes of species with varying capacities for symbiotic nitrogen fixation representing the Cercidoideae and Caesalpinioideae subfamilies were deposited in ArrayExpress under accession E-MTAB-11756.

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
