# Peer review of "Gain or Loss? Evidence for Legume Predisposition to Symbiotic Interactions with Rhizobia via Loss of Pathogen-Resistance-Related Gene Families"

_ijms, 2022, doi:10.3390/ijms232416003_

Round 1

Reviewer 1 Report

The manuscript “ijms-2033088” entitled “Gain or loss? Evidence for legume predisposition to symbiotic interactions with rhizobia via loss of pathogen-resistance related gene families” by Czyż et al. deals with an interesting subject regarding how gene content correlates with the fragmented distribution of nodulation capacity in the Fabaceae family and in the broader spectrum of nitrogen fixing clade (NFC) clade. The results revealed that ten newly sequenced transcriptomes representing species from the Cercidoideae and Caesalpinioideae legume subfamilies with varying nodulation status were supplemented with omics data for an additional 65 angiosperms for orthogroup analysis, enabling categorization of all annotated genes into 64150 orthogroups (i.e. groups of orthologous genes across species). Testing presence/absence and over-representation/under-representation type patterns resulted in the identification of 19 orthogroups with significantly different representation between nodulating versus non-nodulating species. In almost all cases, these orthogroups were more frequently absent in nodulating species. The absence of representation of 13 orthogroups suggested that the losses of their associated genes, including those involved in trichome development, defense, and wounding responses, were strongly associated with rhizobial symbiosis in legumes. This latter finding provides novel evidence of a lineage-specific predisposition for the evolution and/or stabilization of nodulation in Fabaceae, in which a loss of pathogen-resistance genes may have allowed for stable mutualistic interactions with rhizobia.

For publication in the “International Journal of Molecular Sciences”, the topic and content are appropriate. Scientific content and the manuscript size are appropriate. The introduction provides sufficient background and includes all relevant references. In general, the quality of the experiment is well performed and follows rigid scientific logic. The novelty of the results is above the average of novelty knowledge. The conclusions of the article are well-proof. The quality of citations is appropriate. When tables are shown, the units are correctly used. However, there are some points that need attention in order for the article to be published. I would like to recommend the publication of this article, and a minor revision is required for the reasons listed below:

·      Abstract should be rewritten to be more inclusive and within the 200-word limit.

·      Materials and Methods: Statistical analysis sub-section is missing. Please write this sub-section including the experimental layout used in the study and the statistical software package used for the analysis. In addition, please refer to the statistical test used to determine the differences between the parameters (e.g., Student’s t-test).

·      Conclusions: This section is better to be written in one paragraph. A concluding sentence is missing.

·      Lines 661-747: Authors should check and correct the form of references, as in the journal’s “Instructions for authors”.

·      Finally, please carefully revise the back matter section (author contributions, funding, etc.) according to the “Instructions for authors”.

Thank you for your consideration.

Author Response

Answer to the Reviewer 1

We would like to sincerely thank Reviewer 1 for such an in-depth and positive review. We have carefully addressed their suggestions for improvement as follows:

  1. We have edited the abstract to be more inclusive. The changes made resulted in shortening this paragraph to 200 words, which meets the maximum word limit for this section outlined in the author guidelines.
  2. All information regarding statistical testing (i.e. test names, statistical environment) was previously presented in the methods subsection "Identification of gene families differentially represented in nodulating and non-nodulating species". However, to ensure clarity as per the reviewer’s recommendation, we have now split the above-mentioned subsection into two explicit parts, (i) "Identification of gene families across nodulating and non-nodulating species" and (ii) "Statistical testing of differential representation in gene family content". We have additionally added information that statistics module implementing both Fisher exact and Student t tests was scipy.stats.
  3. We have re-written the conclusion paragraph to provide more impact and conclude our research in better way. The final sentence reiterates that not only are our findings of importance for future research on nitrogen fixing symbiotic evolution, but so too are the transcriptomic resources we developed, which will facilitate broader comparative genomics studies in legumes.
  4. Thank you for highlighting the errors in the reference section, as well as in acknowledgments, author contributions and funding sections. We have introduced several corrections to fulfill the requirements of “Instruction for authors” guidelines.

Reviewer 2 Report

The peper can be accepted after a minor revision. The authors could find minor comments in the PDF attached

Author Response

Answer to the Reviewer 2

We would like to thank Reviewer 2 for their feedback. After careful study of the recommendations for taxonomic name formatting by The Legume Phylogeny Working Group (LPWG), we have decided to write the names of the legume family and its subfamilies using normal font, while genus and species names will be written in italics. We also have introduced the required changes in font size/type that were needed to keep sections 2.3 and 2.4 consistent with the remainder of the manuscript.

Reviewer 3 Report

The manuscript titled “Gain or loss? Evidence for legume predisposition to symbiotic 2 interactions with rhizobia via loss of pathogen-resistance related gene families” is devoted to  investigation of how gene content correlates with distribution of nodulation capacity in the Fabaceae family and in the broader spectrum of NFC clade.

Ten newly sequenced transcriptomes representing species from the Cercidoideae and Caesalpinioideae legume subfamilies with varying nodulation status were supplemented with omics data for an additional 65 angiosperms for orthogroup analysis. Testing presence/absence and over-representation/under-representation type patterns resulted in the identification of 19 orthogroups with significantly different representation between nodulating versus non-nodulating species.

This  finding provides novel evidence of a lineage-specific predisposition for the evolution and/or stabilization of nodulation in Fabaceae, in which a loss of pathogen-resistance genes may have allowed for stable mutualistic interactions with rhizobia.

Gene gains (e.g. through ancestral WGDs) do not appear to have directly influenced the origin(s) or generally hypothesized predisposition for symbiotic interaction within the NFC.

This is well-planned and well-done research and high-quality written manuscript. It can be called an exciting reading for any plant scientist, including the Reviewer.

It can be published at present for after additional check for some mis-typings and errors like at Page 5 in the Table 3. Gene families differentially represented in nodulating and non nodulatin(g?) species.

Author Response

Answer to the Reviewer 3

We would like to thank Reviewer 3 for their review and express how privileged we feel to read such a nice opinion on our research and the manuscript itself. We have proofread our work thoroughly once more checking for spelling and syntax mistakes, and have corrected several minor mistakes throughout the manuscript.